# Unknown Slope-Oriented Research on Model Predictive Control for Quadruped Robot

**Zhitong Zhang** **, Honglei An \*, Xiaojian Wei and Hongxu Ma**

College of Intelligence Science and Technology, National University of Defense Technology, Changsha 410073, China
* Correspondence: eric_nudt@163.com; Tel.: +86-17680158919

**Abstract:** There are many undulating terrains in the wild environment. In order to realize the adaptive and stable walking of quadruped robots on unknown sloped terrain, a slope-adaptability model predictive control (SAMPC) algorithm is proposed in this work. In the absence of external vision sensors, the orientation and inclination of the slope are estimated based on the joint position sensors and inertial measurement units (IMU). In an effort to increase the stability margin, the adaptive algorithm adjusts the attitude angle and the touch-down point of the swing leg. To reduce the slipping risk, a nonlinear control law is designed to determine the friction factor of the friction cone constraint from the inclination of the slope. We validate the effectiveness of the framework through a series of simulations. The automatic smooth transition from the flat to the unknown slope is achieved, and the robot is capable of walking in all directions on the slope. Notably, with reference to the climbing modal of blue sheep, the robot successfully climbed a 42.4° slope, proving the ultimate ability of the proposed framework.

**Keywords:** unknown slope; slope estimation; adaptive; model predictive control

## 1. Introduction

The legged robot has the ability to freely select discrete footholds, thus exhibiting better environmental adaptability compared to traditional wheeled and tracked ground-based unmanned platforms. Among them, the quadruped robot combines high stability and fast maneuverability and has attracted high research enthusiasm from the robotics community.

The wild quadrupeds have outstanding performance in terms of high-speed running, weight-bearing, and challenging terrain traversing. The fastest speed of a cheetah can reach over 115 km/h, the 30-cm-long African Goliath frog can jump 5 m, and the blue sheep are good at climbing and jumping, especially on cliffs, as shown in Figure 1a. The locomotor capabilities of these creatures have inspired researchers to study quadruped robots and provide estimable references for the design and control of bionic quadrupeds. It is hoped that quadruped robots will surpass quadrupeds and be able to perform complex tasks in challenging environments. These visions put higher demands on mechanical design, control, perception, and planning.

Since the 21st century, many excellent quadruped robots have emerged at home and abroad, such as Boston Dynamics' Bigdog [1], MIT's cheetah [2], ETH Zurich's ANYmal [3], Unitree A1 [4], and Zhejiang University's Jueying [5]. The problem targeted by the control has also shifted from static position planning [6] to highly dynamic trajectory optimization [7,8]. Nowadays, model predictive control (MPC) has become the most widely used control algorithm in the field of robotics.

Grandia utilized the differential dynamic programming to generate a feedback policy to compensate for the model's mistake [9]; a frequency-dependent cost function was established to reduce the algorithm's sensitivity to high-frequency modeling errors and actuator bandwidth limitations. Bledt proposed a policy-regularized MPC [10] algorithm

that uses several heuristic strategies to generate reference trajectories to achieve stable walking with different gaits. Ding proposed a representation-free MPC framework [11]; the rotation matrix is utilized to directly represent the rotation dynamics, thus avoiding the singularity of Euler angles. The impressive singularity involving a backflip motion demonstrated its effectiveness. Liu analyzed the energy consumption of forward motion about the speed and the height of the torso centroid, and a gait parameter design method was proposed to adaptively select the parameters based on the current speed with minimum energy consumption [12]. Chang proposed a proportional differential (PD) MPC controller to provide compensating forces and moments for the linear motion and rotational motion of the robot, respectively [13], which can compensate for the large modeling errors or unknown loads. Carius combined the MPC with a learning approach and presented a policy-guided search method to train the neural network controller to learn the control laws of MPC [14], which significantly reduces the computational requirements of the controller without losing control performance. Although the above works extend and improve MPC for more flexible and efficient deployment, it can only adapt to slightly uneven ground without any ground information.

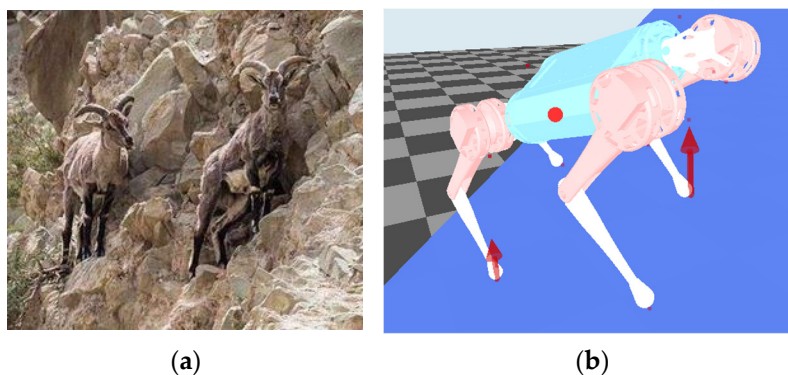

(**a**)    (**b**)

**Figure 1.** The uphill posture of blue sheep and the quadruped robot (under our adaptive algorithm). (**a**) Walking posture of blue sheep; (**b**) Walking posture under SAMPC.

A slope is one of the most typical environments on earth, the stability of locomotion on a slope is a momentous part of the robot's environmental adaptability. Combining the visual, the robot can safely pass through complex terrain through online planning and control [15–17]. In this work, we focus on slope adaptability without robot vision. Liang achieved a 14° slope walking on a physical quadruped by designing a virtual slope to adaptively adjust the torso posture [18]. Zhang used Kalman filtering to estimate the slope angle and uphill direction [19]; the estimation error on a 5° slope is less than 0.2°. Based on the central pattern generator, Xie elevated the touch-down point through the flexion reflex of the knee joint, modified the posture through the vestibular reflex, and finally realized the transition from flat to slope [20]. Liu designed a bionic rigid-flexible coupled quadruped robot based on the motion characteristics of goats [21], and a dynamic center of mass (CoM) adjustment device was designed based on the center of gravity adjustment of goats under different slopes; the maximum slope is 30°. Liu extracted the biological characteristics of sheep's hoof and designed a bionic anti-slip hoof based on the tension principle, it was found that the bionic hoof with a low hardness sole and large elastic coefficient ligament has the best slip resistance [22]. Yu integrated the posture correction controller into the foot trajectory generator [23]. A Proportion Integration Differentiation (PID) controller is used to adjust the body posture to reduce the variance degree of the body when the robot upward or downward 10° slope. This method may help the transport task, but it is not conducive to walking on large slopes. Focchi proposed a control framework for quasi-static walking [24]; the quadratic programming (QP)-based force distribution method results in no foot slippage and minimization of the actuator's effort, although they realized that in high-slope (50°) walking, the V-shaped wall terrain is uncommon and the quasi-static gait

is not suitable for fast walking. The above methods achieve basic walking on slope terrain based on bionic mechanism analysis or Zero Moment Point (ZMP) theory, and there is still room for further improvement in slope adaptability.

In order to improve the adaptability of the quadruped robot to unknown slope terrains, a slope adaptability model predictive control (SAMPC) is proposed, as shown in Figure 2. The operator inputs the desired motion commands, including forward velocity $v_x^d$, lateral velocity $v_y^d$ and yaw angular velocity $\dot{\psi}^d$ through the joystick; the reference trajectory planner integrates these commands to obtain the reference trajectory $x_{ref} \in \mathbb{R}^{12}$ of the CoM within the prediction horizon. The MPC controller optimizes the feedforward force $f_{ff} \in \mathbb{R}^{12}$ and plans the movement of the stance leg in the hip frame, $P_{st}^{ref}, V_{st}^{ref} \in \mathbb{R}^3$ are position and velocity vector of the foot respectively. The swing planner plans the movement of the swing leg to reach the next touch-down point. The low-level joint controller tracks these control commands to generate the joint driving torque $\tau \in \mathbb{R}^{12}$. $q, \dot{q} \in \mathbb{R}^{12}$ are the joint angle and angular velocity. The state estimator estimates the robot's state $\hat{x}$ according to the information of the proprioception. In the absence of external sensors, the slope-adaptive algorithm used to estimate the orientation $\theta_n$ and the inclination $\psi_n$ of the unknown slope, and a series of adjustment methods for the other parts of the control framework is established based on the estimated information, which are the blue parts in Figure 2.

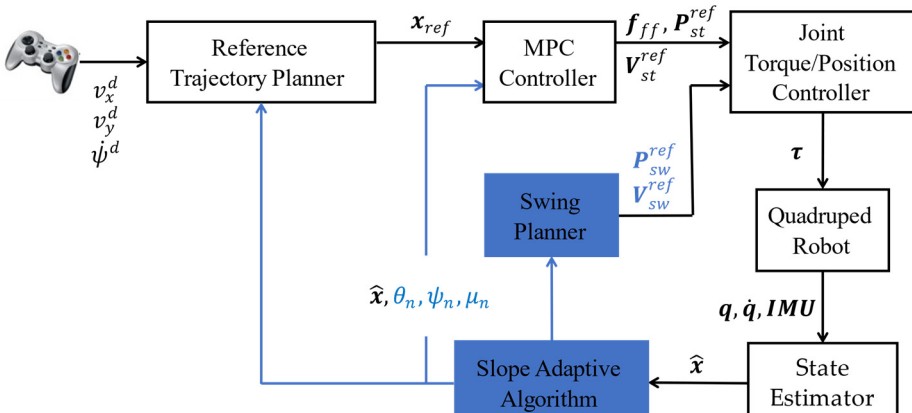

**Figure 2.** Control framework of the adaptive MPC for an unknown slope.

Simulation experiments verified the effectiveness of the algorithm, and achieved a smooth transition when the robot walked from flat to slope. Finally, referring to the mode of blue sheep uphill, our robot successfully topped the 42.4° slope, which approached the physical limit. The remainder of this paper is organized as follows. Section 2 briefly presents the visionless motion control algorithm of MPC. Section 3 introduces the slope-adaptive algorithm and contains slope estimation and adaptive adjustment of SAMPC. The validation experiments of the simulation platform and the analysis of the results are illustrated in Section 4. Finally, Section 5 concludes the whole work and proposes future work.

## 2. MPC Formulation

Model predictive control is a classic model-based method, which uses a mathematical model to represent the kinematic and dynamical properties of the control object, and the model is used to predict the future state in a finite prediction horizon based on the current state and control input. The optimal control considers various explicit constraints, and solves the control input to minimize the cost function regarding the desired task; only the first step solution is applied to the object. This receding horizontal optimization is able to handle the disturbances caused by internal or external uncertainties so as to achieve stable control.

### 2.1. Simplified Dynamic Model

In our application, the quadruped robot model and the coordinate system are shown in Figure 3. The inertial frame and floating base frame are recorded as $O_I$ and $O_b$ respectively. $\varphi$, $\theta$ and $\psi$ respectively represent the pitch angle, roll angle and yaw angle of the body relative to the inertial frame, which form the Euler angle $\boldsymbol{\Theta} = [\varphi, \theta, \psi]^T$ according to the rotation order of Z-Y-X. $\boldsymbol{f}_i \in \mathbb{R}^3$ and $\boldsymbol{r}_i \in \mathbb{R}^3$ are the foot reaction force and the position vector from the foot end to the CoM in the inertial frame, the subscript $i$ represents the indicia of leg.

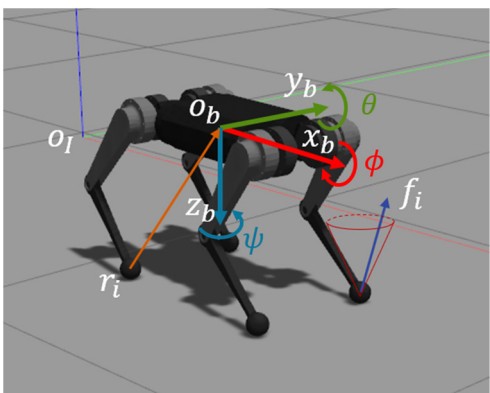

**Figure 3.** Schematic diagram of the robot and coordinate system.

Each leg contains three joints, which are called side-hip, front-hip, and knee, respectively. Furthermore, the three links of the leg are called abduction-adduction (abad), thigh, and calf, respectively. All motors are mounted compactly around the trunk and integrated into a virtual body. The two hip joints are directly driven, and the knee joint is driven by a connecting rod hidden in the thigh.

Quadruped robots have the characteristics of high dimensionality and strong nonlinearity, making it difficult to guarantee the real-time property of the receding horizontal optimization. The complete state space consists of 6-dimensional posture, 12-dimensional joint angle and and their first-order differential, the total 36-dimensional state space is not conducive to the optimal calculation and is difficult to ensure real-time performance.

Generally, under reasonable assumptions, the model can be simplified into a linear model to improve the speed of optimization and reduce the computing requirements of an onboard computer. Compared to the BigDog [1] with hydraulic actuators mounted on the legs and the ANYmal [3] with knee motors mounted on the thigh, the leg's mass is much smaller than the virtual body's mass, so the influence of the leg's movement on the body can be ignored. Therefore, the whole body dynamics can be simplified to a single rigid body model with four massless variable-length rods. According to the Newton-Euler equation, the simplified dynamic model is obtained as follows:

$$\ddot{\boldsymbol{p}} = \frac{\sum_{i=1}^{4} \boldsymbol{f}_i}{m} + \boldsymbol{a}_g \tag{1}$$

$$\frac{d}{dt}(\boldsymbol{I}\boldsymbol{\omega}) = \boldsymbol{I}\dot{\boldsymbol{\omega}} + \boldsymbol{\omega} \times \boldsymbol{I}\boldsymbol{\omega} = \sum_{i=1}^{4}(\boldsymbol{r}_i \times \boldsymbol{f}_i) \tag{2}$$

where $\boldsymbol{p}, \boldsymbol{\omega} \in \mathbb{R}^3$ are the position and angular velocity of the body, $m, \boldsymbol{I} \in \mathbb{R}^{3\times3}$ are the mass and the inertia matrix of the virtual body, $\boldsymbol{a}_g \in \mathbb{R}^3$ represents the gravitational acceleration vector. Assuming that the deviation of attitude is relatively small during horizontal motion, the gyro moment term $\boldsymbol{\omega} \times \boldsymbol{I}\boldsymbol{\omega}$ in Equation (2) can be neglected. In order to form a standard

equation of state (EOS), we take $a_g$ as an additional state variable, and the simplified EOS is obtained:

$$
\begin{bmatrix} \dot{\theta} \\ \dot{\omega} \\ \dot{p} \\ \ddot{p} \\ \dot{a}_g \end{bmatrix} = \begin{bmatrix} 0_3 & R^T & 0_3 & 0_3 & 0_3 \\ 0_3 & 0_3 & 0_3 & 0_3 & 0_3 \\ 0_3 & 0_3 & 0_3 & 1_3 & 0_3 \\ 0_3 & 0_3 & 0_3 & 0_3 & 1_3 \\ 0_3 & 0_3 & 0_3 & 0_3 & 0_3 \end{bmatrix} \begin{bmatrix} \theta \\ \omega \\ p \\ \dot{p} \\ a_g \end{bmatrix} + \begin{bmatrix} 0_3 & \cdots & 0_3 \\ 0_3 & \cdots & 0_3 \\ I^{-1}r_1 & \cdots & I^{-1}r_4 \\ 1_3/m & \ldots & 1_3/m \\ 0_3 & \cdots & 0_3 \end{bmatrix} \begin{bmatrix} f_1 \\ f_2 \\ f_3 \\ f_4 \end{bmatrix} \tag{3}
$$

where $R \in \mathbb{R}^{3\times3}$ is the rotation matrix of the body frame $O_b$ in the inertial frame $O_I$, $0_3$ and $1_3$ are diagonal matrices. Equation (3) is discretized by the zero-order hold method to obtain Equation (5), which is the prediction model for MPC.

*2.2. Receding Horizontal Optimization*

For the robot with horizontal locomotion, the desired movement commands are $v_x^d$, $v_y^d$, and $\dot{\psi}^d$. The trajectory planner generates the reference trajectory of CoM $x_{ref} \in \mathbb{R}^{13}$ within prediction horizon $N$ based on the commands and the current states. In order to track the reference trajectory, an optimization problem is established to find the optimal control sequence, i.e., the desired foot reaction force during the stance phase to drive the body.

$$
\min_u J(u) = \sum_{n=0}^{N-1} ||x_{n+1} - x_{n+1,ref}||_{Q_x} + ||u_n||_{R_u} \tag{4}
$$

$$
s.t. : \quad x_{n+1} = A_n x_n + B_n u_n \tag{5}
$$

$$
D_n u_n = 0 \tag{6}
$$

$$
\underline{c}_n \leq C_n u_n \leq \overline{c}_n \tag{7}
$$

where $n$ is the control horizon of MPC, the optimization variables $u_n = [f_1^n, f_2^n, f_3^n, f_4^n]$ are the foot force at $n$th step. $J(u)$ is the optimization objective function, the first term aims to minimize the trajectory tracking error, and the second term is to minimize the control effort, $Q_x \in \mathbb{R}^{13\times13}$ and $R_u \in \mathbb{R}^{12\times12}$ are the corresponding weight matrix, which trades off all aspects in the process of optimization.

The predictive model Equation (5) is used to deduce the dynamics of the robot based on the given control input, $A_n \in \mathbb{R}^{13\times13}$ is the state transition matrix, $B_n \in \mathbb{R}^{13\times12}$ is the control matrix. Equation (6) contains gait information, $0 \in \mathbb{R}^{12\times1}$ is a zero vector, $D_n = diag(d_1^n, d_2^n, d_3^n, d_4^n)$ is the force selection matrix, $diag()$ represents the diagonal matrix. $d_i^n \in [\mathbb{O}_3, \mathbb{I}_3]$, $\mathbb{O}_3$, $\mathbb{I}_3 \in \mathbb{R}^{3\times3}$ are diagonal matrices and the entry equals 0, 1, respectively. This equation constrains the availability of optimization variables. The submatrix is set according to the phase of each leg, when a leg is in the stance phase, the corresponding $d_i^n$ is set to $\mathbb{O}_3$, so the the value of corresponding $f_i^n$ is unconstrained. On the contrary, the subdiagonal matrix of the swing leg is set to $\mathbb{I}_3$, so the corresponding optimization variables are constrained to zero; this conforms to the fact that there is no contact between the swing leg and the ground.

Inequality Equation (7) expresses the constraint relationship of each component of foot force, and it can be decomposed into the following two equations:

$$
0 \leq f_z \leq \overline{f} \tag{8}
$$

$$
-\mu f_z \leq \pm f_{x,y} \leq \mu f_z \tag{9}
$$

Considering the torque limitation of the actual joint motor, Equation (8) limits the vertical force component $f_z$. In order to avoid foot slipping, Equation (9) requires the foot force to satisfy the friction cone constraint, where $\mu$ is the friction factor of foot-ground contact, and $f_x$ and $f_y$ are the forward component and lateral component, respectively.

Although the above optimization problem solves *N* steps control inputs, the long-term control sequence cannot meet new control requirements because of the unknown disturbance caused by the uneven environment or external force. In order to improve the adaptability and response quickly to the deviated state, only the first step of the optimal control sequence $u_0 = \left[ f_1^0, f_2^0, f_3^0, f_4^0 \right]^T$ will be sent to the leg controller, and the optimization will be repeated before the next MPC step comes; this is known as receding horizontal optimization.

### *2.3. Leg Controller*
#### 2.3.1. Stance Leg

The quadruped robot is an intermittent underactuated system. The stance legs are in charge of the whole motion; for the trot gait, the support line formed by two diagonal stance legs is insufficient to provide the torque around it. MPC provides a certain predictive ability for this underactuated condition to obtain effective stability, the same for the underactuated condition that all four legs are in the swing phase.

The low-level controller is established in the hip frame of each leg, the hip frame is fixed with the body, located at the abad joint. Since the optimization algorithm is defined in the inertial frame, the expected foot reaction force $f_i^0$ need to convert to the hip frame as the feedforward foot force:

$$f_{ff} = -R^T \cdot u_0 \tag{10}$$

Assuming that the stance feet do not slide during the stance phase, the reference motion of the CoM can be mapped to the motion of the foot end $P_{st}^{ref}$ and $\dot{P}_{st}^{ref}$ under the hip frame; this track will also be controlled.

#### 2.3.2. Swing Leg

The control of the swing leg includes two parts: the touch-down point and trajectory. The trajectory starts from the position where the foot lifts off and ends with the touchdown point. For the plane walk robot, the expected touch-down point of the swing leg is designed in the hip frame:

$$P_{sw}^{ref} = \frac{v_{com} T_{st}}{2} + k_v \left( v_{com} - v^{ref} \right) \tag{11}$$

where $v_{com} \in \mathbb{R}^2$ is the current horizontal velocity of the robot, and $T_{st}$ is the predefined supporting duration. The first item of Equation (11) represents a half-step length, for uniform motion, the foot trajectory is symmetrical with respect to the hip. The second one is for motion compensation, $k_v \in \mathbb{R}^2$ is the compensation factor used to correct the tracking error of the velocity of CoM.

The Bezier curve is used to generate the foot trajectory between the touch-down point and lift-off point. A suitable vertical height is set for clearance of the ground obstacle, and several control points are used to smooth the path, arrange the desirable swing leg retraction rate [25], and reduce the impact.

Combining the control demands of the stance leg and swing leg, the following joint torque/position controller is obtained:

$$\tau_i = J_i^T \left[ f_{ff} + K_p \left( P_i^{ref} - P_i \right) + K_d \left( \dot{P}_i^{ref} - \dot{P}_i \right) \right] \tag{12}$$

where $\tau_i \in \mathbb{R}^3$ are the joint control torque vector of each leg, $J_i \in \mathbb{R}^{3 \times 3}$ is the Jacobian matrix of *i* leg that maps the force in operational space to the torque in joint space. $K_p, K_d \in \mathbb{R}^3$ are the proportional coefficient and damping coefficient of the position control in operating space, respectively. It is worth noting that the position control has different functions in different phases. The swing leg needs to track the desired trajectory with higher accuracy; the coefficient is larger, and the stiffness is higher. However, in the support phase with

optimal feedforward force, the position control is only used to compensate for the tracking of the CoM for the reference trajectory, so the coefficient is smaller and the stiffness is lower.

## 3. Slope-Adaptive Algorithm

The MPC algorithm in Section 2 has good performance on level ground or slightly uneven terrain. However, due to the time-based gait schedule, the force selection matrix is determined only by time and gait patterns. Without terrain awareness, a robot cannot adapt to complex terrains such as slopes. When the robot walks on a slope, the actual touch-down event is bound to be different from the planning; the swinging leg touches the ground in advance still using the high stiffness position control, which will affect the stability of the robot. The higher the inclination, the stronger the impact.

For the above disturbances, the compensation ability to recede horizontal optimization is no longer effective, and the estimation of the orientation and inclination of an unknown slope can essentially solve this problem.

### 3.1. Slope Estimation

When the robot enters the slope with a trot gait, the diagonal legs will be in different planes, as shown in Figure 4.

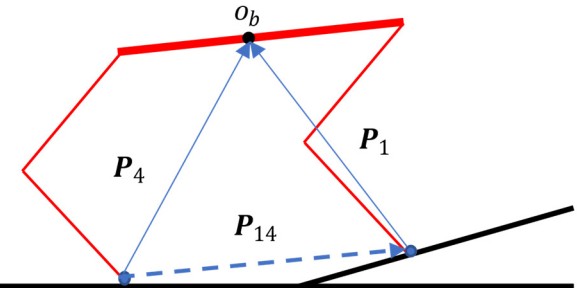

**Figure 4.** Schematic diagram of quadruped slope walking.

The 1–4 diagonal legs and body form a space link closed chain. Based on forward kinematics, the foot position vector $P_1$ and $P_4$ under the body frame is obtained according to the joint angle sensors. Combined with the body attitude estimated by IMU, the position vectors of the two touch-down feet in the inertial frame can be obtained:

$$P_{14} = R(P_1 - P_4) \tag{13}$$

Thus, according to this vector, the nominal pitch angle $\theta_n$ can be estimated based on the components of $P_{14}$. Moreover, according to the forward rotation direction defined by the right-hand system, a minus sign is required:

$$\theta_n = -atan(P_{14}^z / P_{14}^x) \tag{14}$$

However, because the slope direction is unknown, the robot's uphill direction does not necessarily coincide with the main slope direction, as shown in Figure 5.

When the robot ascends the slope head-on, as the left robot in Figure 5, the estimated nominal pitch angle is approximate to the real inclination $\theta_s$ of the slope. When the robot deflects from the slope, as in the right robot in Figure 5, the estimated nominal pitch angle is unequal to the inclination. Therefore, the correct estimation also needs to know the orientation angle of the slope.

Due to the deflection angle between the robot and the slope, the two front legs do not enter the slope at the same gait cycle; there is an altitude difference between the two

touch-down points, then it can be judged that the uphill direction at this time is not the slope orientation, so a heuristic algorithm is designed:

$$\dot{\psi}^{ref} = k_\psi P_{12}^z \tag{15}$$

where $k_\psi$ is the parameter of adjustment steering speed. $P_{12}^z$ is the vertical direction component of the position vector from 2 legs to 1 leg. The robot is guided to turn according to the altitude difference. With the robot turning to the slope, the difference gradually decreases, decreasing the expected steering speed. When the movement reaches the steady state, the current robot yaw angle is regarded as the slope orientation $\psi_s$. Moreover, Equation (13) is replaced by the following formula; only the modified position vector contains the true slope information.

$$P_{14} = R_{-\psi_s} R(P_1 - P_4) \tag{16}$$

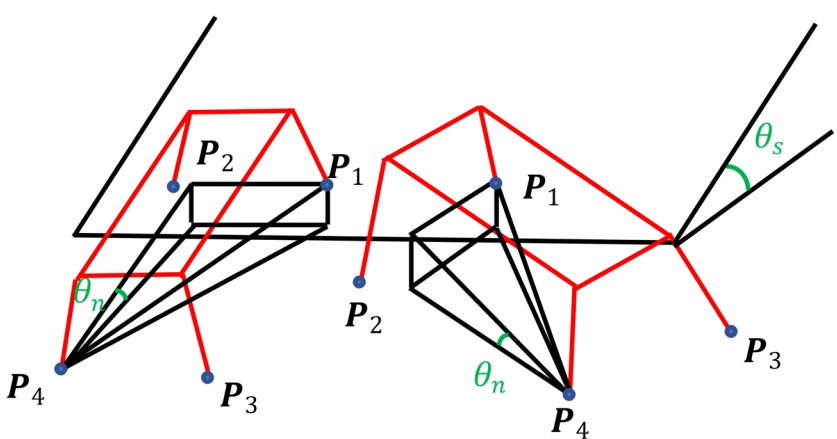

**Figure 5.** Entering the slope with different directions.

*3.2. Adaptive Adjustment*

3.2.1. Desired Attitude Angle

The goat will adjust its body to keep parallel to the slope, and this climbing posture provides higher stability [18]. Referring to blue sheep, the desired attitude angle can be adjusted according to the estimated slope information. By the way, in order to enable the robot to turn at will on the slope and walk stably in all directions, it is necessary to assign the estimated nominal inclination to the desired pitch angle $\theta^{ref}$ and desired roll angle $\varphi^{ref}$ according to the actual yaw angle:

$$\theta^{ref} = \theta_n \cdot cos(\psi - \psi_s) \tag{17}$$

$$\varphi^{ref} = \theta_n \cdot sin(\psi - \psi_s) \tag{18}$$

3.2.2. Touch-Down Point

Based on the new desired attitude angle, the original horizontal reference trajectory needs to be updated along the uphill direction. The speed in the original horizontal direction should be converted to the slope direction, and the reference trajectory of CoM also changed accordingly.

$$v_{world}^{ref} = R \cdot v^{ref} \tag{19}$$

For the touch-down point planning on a slope, Equation (11) is insufficient. The new desired attitude makes the body always parallel to the slope. As shown in Figure 6, the cross symbol represents the CoM of the virtual body. the red robot has not undergone any adjustment; when it is in the middle stage of the stance phase, the projection of CoM is in the middle of the support line. For the green one, after posture adjustment, due to the local

planning of foot position in the hip frame, the support line moves up because it does not take into account the pitch angle of the body; the projection of CoM is closer to the rear foot, so the stability is lower based on the ZMP theory.

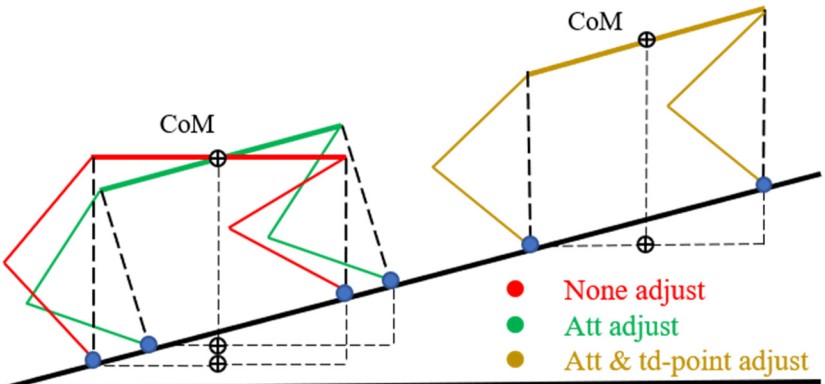

**Figure 6.** The adjustment of foot touch-down point.

According to the current posture of the robot, the yellow robot plans in the inertial frame so that the foot is directly beneath the hip on the slope. Equation (11) is calculated under the inertial frame and mapped back to the hip frame through the rotation matrix.

$$\boldsymbol{P}_{sw}^{ref} = \boldsymbol{R}^T \left( \boldsymbol{P}_{hip}^{ref} + {}^I\boldsymbol{P}_{sw}^{ref} - H \right) \tag{20}$$

where ${}^I\boldsymbol{P}_{hip}^{ref}$ and ${}^I\boldsymbol{P}_{sw}^{ref}$ are the position of each hip frame and expected touch-down point in the inertial frame, $H$ is the virtual leg length. The adjusted touch-down point makes the robot still maintains a strong stability margin on the inclined plane.

### 3.2.3. Friction-Cone Constraint

For all ground robots, the friction characteristics of the contact part determine the performance of the motion [26]. It is essential to prevent lack of control caused by slipping during locomotion. For the slope motion, the friction force decreases as the slope increases, and the dynamic friction factor required for self-locking is increased.

In the framework of MPC, the control of slipping is embodied in Equation (9). The predetermined parameter $\mu$ for flat ground is not applicable to slope movement; it is easier to slip due to the reduction of normal pressure. Therefore, it is necessary to adjust the range of the friction cone according to the slope:

$$\mu_n = 0.5 \times exp(-\theta_n \cdot \beta) + 0.1 \tag{21}$$

where $\beta$ is the regulatory factor. The updated factor $\mu_n$ decreases with the slope increase. A smaller factor constraint MPC generates the desired foot force more strictly, making the resultant force closer to the normal vector of the slope and reducing the risk of slipping. The nonlinear form of Equation (21) decreases the factor rapidly when the slope is small, which helps the robot make a smooth transition to the slope.

Based on Sections 2 and 3, the framework of the adaptive model predictive control algorithm for unknown slope is established, as shown in Figure 2, where the blue part is the innovative work of this paper.

## 4. Results and Discussion

This section verifies the proposed unknown slope-adaptive model predictive control algorithm. In the simulation environment, the friction factor is set to 1, and the angle and direction of the slope can be set at will. The optimization problem of Equations (4)–(9) is

solved by the open-source qpOASES Slover. The robot model parameters and simulation parameters are shown in Table 1:

**Table 1.** Model parameters and simulation parameters.

| Parameters | Value | Unit |
|:---:|:---:|:---:|
| $m$ | 9 | kg |
| $I$ | $diag(0.112, 0.362, 0.426)$ | kg·m² |
| body length $L$ | 0.38 | m |
| body wide $B$ | 0.1 | m |
| link length $l$ | (0.072, 0.2, 0.2) | m |
| swing leg $K_p$ | $diag(700, 700, 150)$ | |
| swing leg $K_d$ | $diag(7, 7, 7)$ | |
| stance leg $K_p$ | $diag(50, 50, 10)$ | |
| stanceleg $K_d$ | $diag(2, 2, 2)$ | |
| $N$ | 10 | |
| $n$ | 15 | |
| step of leg controller $T_{step}$ | 2 | ms |
| half gait cycle $T_{trot}$ | 300 | ms |
| $Q_x$ | $diag\begin{pmatrix} 1.25, & 1.25, & 10, & 2, & 2, & 50 \\ 0, & 0, & 0.3, & 1.5, & 1.5, & 0.2, 0 \end{pmatrix}$ | |
| $R_u$ | $0.00004 \times \mathbb{I}_{12}$ | |
| $\beta$ | 4 | |

The control horizon of MPC $n = 15T_{step}$, which means the low-level leg controller executes the same MPC feedforward force within 30 ms. Moreover, the prediction horizon $N = 10n$, which is equal to the half-trot gait cycle, means that with this constraint, a foot will not touch down the ground twice in one prediction horizon. Figure 7 shows the speed tracking performance of the original MPC controller in plane motion, the red lines and the blue lines represent the forward velocity and the lateral velocity respectively, the dotted line and the solid line represent the reference velocity and the actual velocity respectively. The average tracking error in both directions is less than 0.01 m/s, the mean square error (MSE) is 0.0019 and 0.0017 respectively.

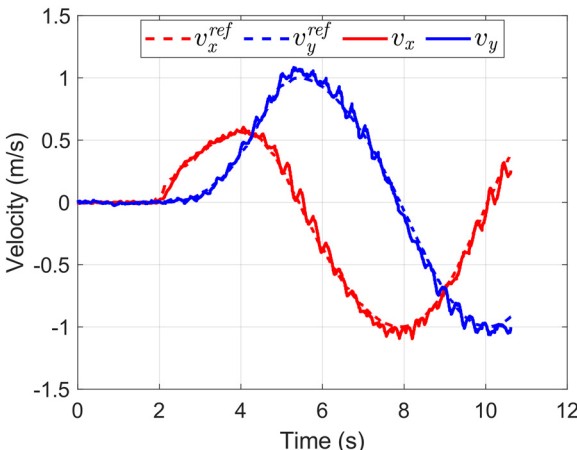

**Figure 7.** Horizontal locomotion under original MPC.

### 4.1. Comparison of the Posture

Although the original controller has good tracking performance for the commands on flat ground, it has no adaptability to the slope, as shown in Figure 8a. The slope is set to 30°; the robot is still expected to keep the body level during the uphill process, resulting in the front leg being squeezed and the back leg stretched, which is not conducive to the movement of the slope. The posture under SAMPC is shown in Figure 8b,c; whether

forward or lateral uphill, the adaptive algorithm keeps the body parallel to the slope, the virtual leg length of each leg remains the same, and the adjusted foot touch-down points keep the CoM in the center of the ZMP stable region.

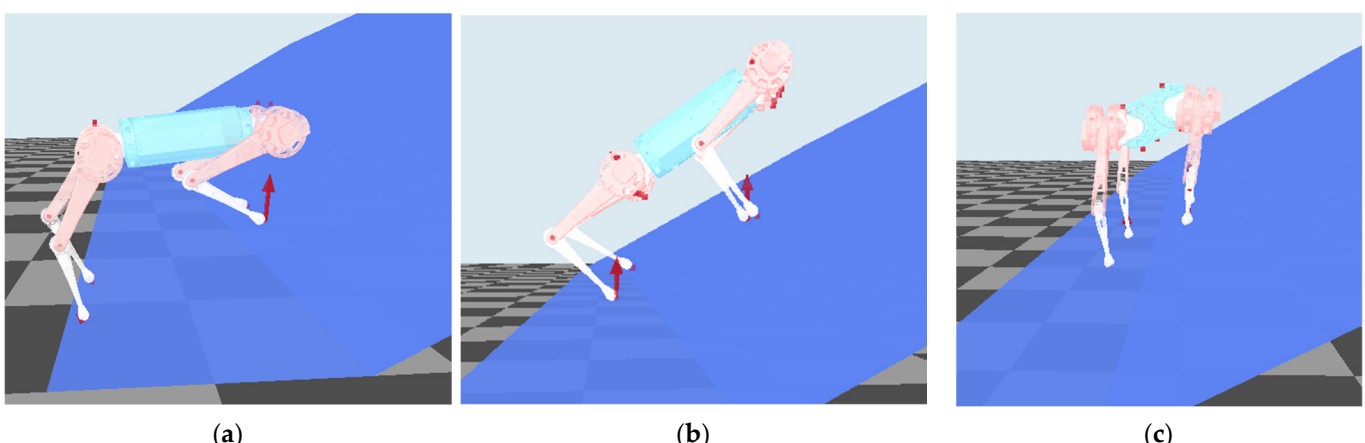

**Figure 8.** Comparison of the slope walking posture. (**a**) Before adjustment; (**b**) Forward walking after adjustment; (**c**) Lateral walking after adjustment.

### 4.2. Slope Estimation and Adjustment

An unknown slope with a 30° inclination and 30° deflection angle is placed at 1 m forward of the robot. As shown in Figure 9a, the robot first moves forward and then enters the slope at 1.6 s, the gradient of position changes accordingly. Figure 9b shows the estimation of the slope and the adjustment of the friction factor. After 3 s, the slope enters the steady state, with a steady state estimation error of 6.7%, and the estimation of slope orientation is more accurate. For the convenience of explicit expression, the friction factor is magnified by 50 times, gradually decreasing from 0.6 to 0.15.

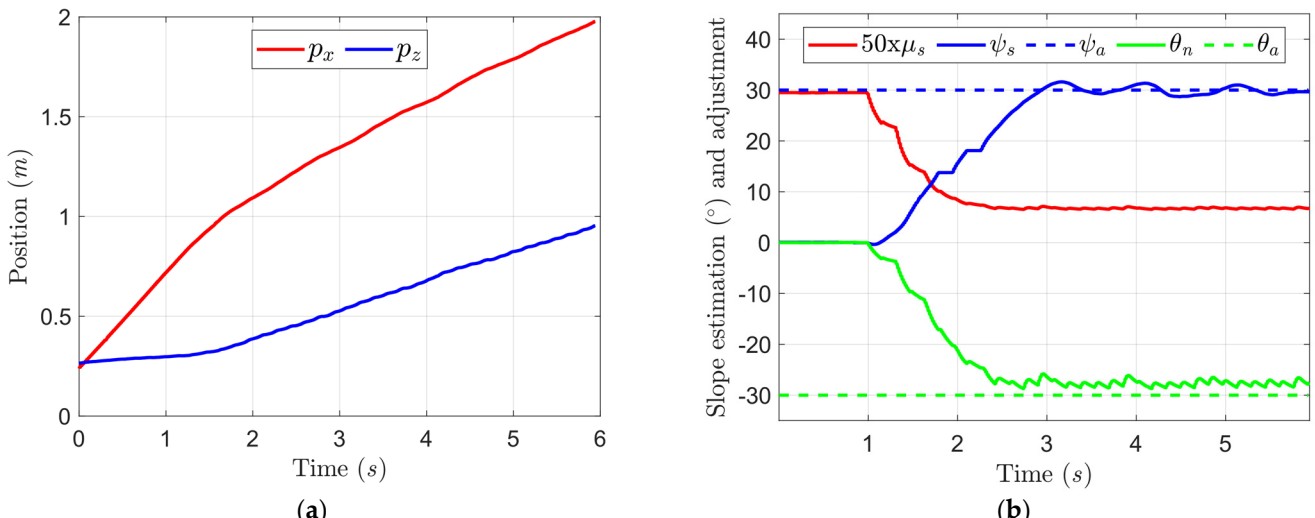

**Figure 9.** Uphill process of the unknown slope. (**a**) CoM position on a sagittal plane; (**b**) Estimation and adjustment.

Figure 10 shows the estimation of different unknown slopes, the Box-plot is used to describe the statistical characteristics of the estimation, the black bar indicates the extreme value, the red bar indicates the median values, the blue box connects upper and lower quartiles, the shorter the Box-plot, the more concentrated the estimation. Six slopes are set from 0.1 rad to 0.6 rad. It can be seen in Figure 10a, with the increase of the slope, a

sight slip will occur, so the steady state error and oscillation amplitude of the estimation increase gradually.

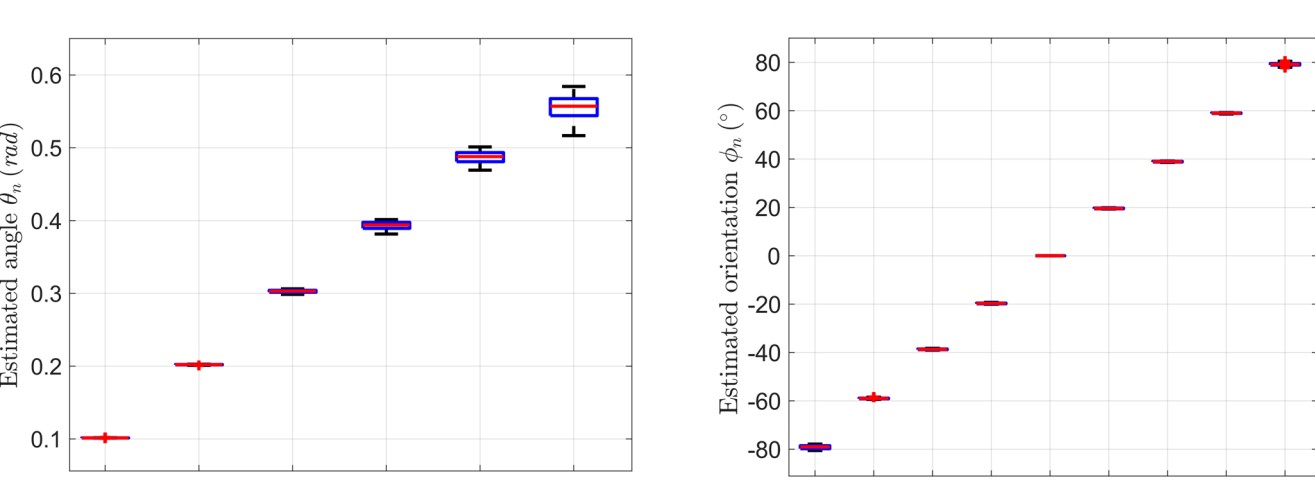

(**a**)                    (**b**)

**Figure 10.** Estimation of different slopes. (**a**) The estimation of slope inclination; (**b**) The estimation of slope orientation.

For the estimation of slope orientation, the initial yaw angle of the robot is set to $0°$. Slopes with $30°$ inclination are divided into 9 groups based on the deflection angle within the range of $90°$, as shown in Figure 10b; since the robot turns from an initial yaw angle to a larger slope orientation angle, the altitude difference gradually diminish and finally ignored, so the estimated angle is always less than the actual value, the estimation error is less than is within $3°$ anyway.

### 4.3. Omnidirectional Walking and Extreme Climbing

In order to execute various tasks, robots need to have omnidirectional walking ability on slopes. Figure 11 shows the omnidirectional walking process of the robot on a $35°$ slope. The desired yaw angular velocity is fixed, the robot rotates in a reverse direction, and the actual yaw angle is shown as the black line; the instantaneous hop is due to the yaw angle defined in $\pm\pi$. The estimated nominal pitch is adaptively assigned to two desired attitudes, pitch, and roll, based on the current yaw, so the trunk is always parallel to the slope.

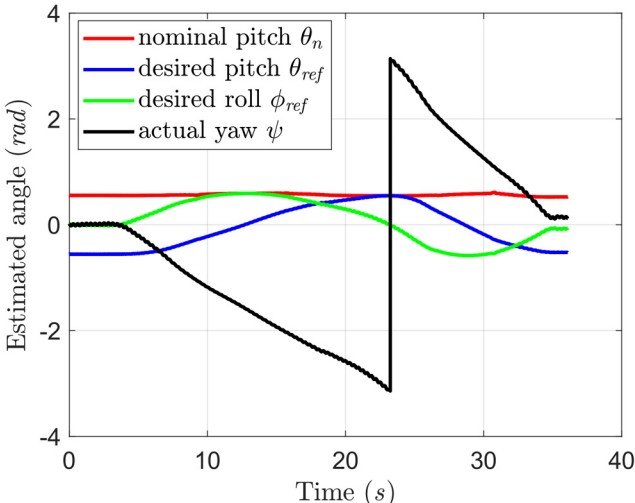

**Figure 11.** Omnidirectional walking on a $35°$ slope.

The self-locking angle reflects the contact properties between different materials and depends on the friction factor. When the slope angle is less than the self-locking angle, the static object on the slope will not slide, that is, a self-locking phenomenon, for the moving objects only powered by friction, it is dangerous and slippery to move on a slope with the inclination close to the self-locking angle. In nature, the body width of quadrupeds is far less than the body length, so the blue sheep often adopt the mode of lateral climbing in extremely steep terrains, which is more conducive to leg movement. Similarly, when we ride a bicycle up the slope, we often take an S-shaped route, which can reduce the gradient of the route.

As the friction factor is set to 1, the self-lock angle is 45°. As shown in Figure 12a, there are six snapshots in uphill order; with the reference to blue sheep, the maximum climbing inclination angle under SAMPC is 42.4°. Figure 12b illustrates three robot states uphill; the red line represents the altitude of the CoM of the robot, the green line represents the actual yaw angle, and the blue indicates the estimated nominal pitch angle. Because the slope is too tilted, the front leg skids on the slope, and the rear leg cannot propel the robot up, so the robot switches to a lateral posture at step 2; the steering is completed at 20 s, and then enters the slope smoothly. The gradually increased estimated nominal slope is assigned to the desired roll angle, and returns to zero after reaching the summit.

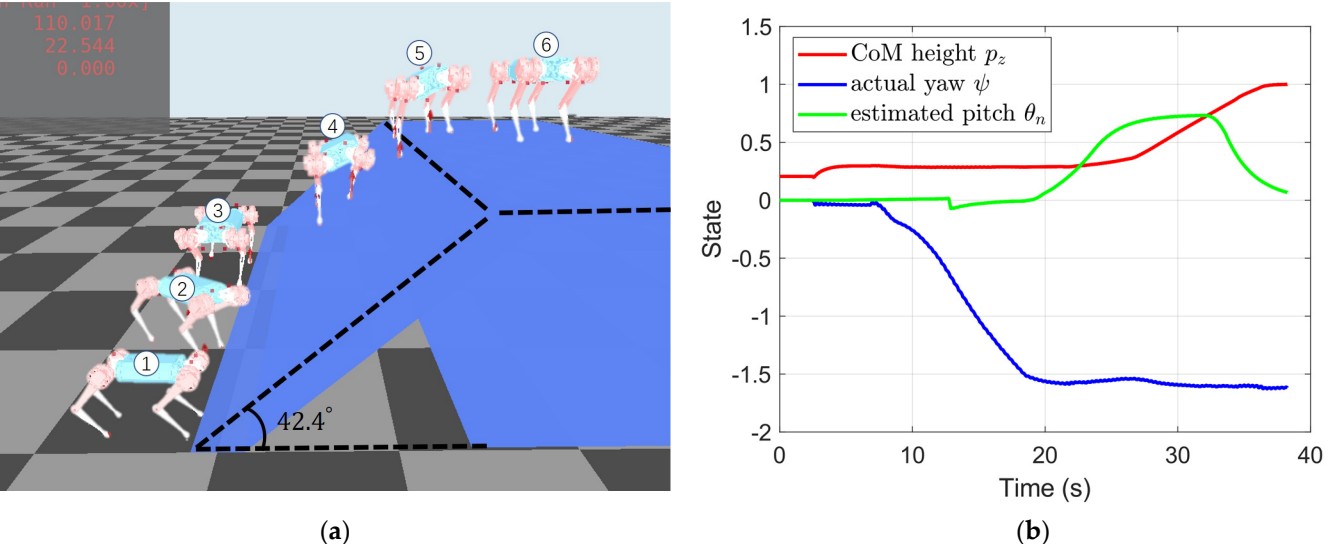

(**a**)       (**b**)

**Figure 12.** Process of extreme climbing. (**a**) Snapshot of the uphill process; (**b**) State of the uphill process.

## 5. Conclusions

In this work, we proposed a slope-adaptive model predictive control (SAMPC) to realize the stable locomotion on an unknown slope for a quadruped robot. By fusing IMU and joint angle sensors, the slope orientation and inclination are estimated, and the adaptive algorithm adjusts the desired attitude so that the robot's body is always parallel to the slope. In order to leave sufficient motion space for the front leg and maintain a large stability margin, the touch-down point has also been adjusted. At last, the adaptive friction factor utilized in the constraint of optimization reduces the risk of the foot slipping by generating a force more perpendicular to the inclined face. Above all, the smooth transition of the robot from the flat to the slope is realized, and the robot is able to walk in all directions on the slope. Finally, referring to the climbing mode of blue sheep, the successful ascent of the 42.4° slope proves the ultimate ability of the SAMPC.

In future work, firstly, we plan to deploy the SAMPC to the physical platform UnitreeA1 to verify the performance further. Secondly, the manually designed adjustment law of friction factor is not the optimal choice definitely; combining the learning-based method, such as utilizing the policy search to choose the optimal traversal time for agile drone flight [27], the optimal factor can be found automatically for different slopes.

As we all know, the slope is just one kind of natural terrain. In order to enable robots to achieve or exceed the environmental adaptability of quadruped mammals, it is necessary to consider the adaptation problems in a variety of complex environments, utilize external sensors such as depth cameras to achieve upper-level motion planning in the limited landing areas, further improving the autonomous interaction ability of robots to the challenging environment. Despite all this, our work is vital for conditions in which the visual sensor is not available, such as heavy fog, strong light, or sensor damage.

**Author Contributions:** Conceptualization, Z.Z. and H.A.; Formal analysis, Z.Z. and H.A.; Investigation, Z.Z. and X.W.; Methodology, Z.Z.; Project administration, H.M.; Resources, H.M.; Software, Z.Z.; Supervision, H.M.; Validation, Z.Z. and H.A.; Writing—original draft, Z.Z.; Writing—review & editing, H.A and X.W. All authors have read and agreed to the published version of the manuscript.

**Funding:** This research received no external funding.

**Data Availability Statement:** Not applicable.

**Conflicts of Interest:** The authors declare no conflict of interest.

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
