# Peer review of "Unknown Slope-Oriented Research on Model Predictive Control for Quadruped Robot"

_machines, doi:10.3390/machines11020133_

Round 1

Reviewer 1 Report

- The sentences used in the article are too long, which makes the article difficult to understand. Authors should reorganize the article with more simple sentences.

- Only the simulation results of the proposed control method are shared. Why the experimental study was not carried out should be explained.

- The results obtained were not compared with the results of studies in the literature using other control methods.

- A sufficient number of references are given and the references given are relevant to the topic.

Reviewer 2 Report

In the manuscript there is proposed a slope adaptive model predictive control (SAMPC) to realize the stable locomotion on unknown slope for quadruped robot. The designed algorithms have been verified by simulation experiments with result of successfully topped the 42.4° slope.

Comments and Questions to the Authors:

1. The units are atypically italicized in the text, in some cases the space between the value and the unit is missing. Please check it.

2. Figures 1-6 are titled as “Figure”, figures 7-12 as “Fig.”. Please unify. 

3. Page 3, Line 86: please correct “uesd”.

4. Page 3, Figure 2: Control framework of the adaptive MPC for unknown slope is already presented in Section 1 Introduction, but without its description. Please add short description with meaning of presented variables. Also please correct “unknow”.

5. Page 4, Lines 133-134: There is written “…m and I ∈ ℝ3×3 133 are the mass and the inertia matrix of virtual body,...“, but they are not in bold, "m" also in Eq. (1).

6. Page 5, Lines 158-159: The sentence “When the leg in the swing period, the corresponding diagonal elements are set to 1, so the optimization variable constrained to 0.“ is a bit difficulty to understand.

7. Page 5. Line 181: The beginning of a sentence “leg control mapping the operation space foot forces to joint space torque, ...“.

8. Page 6. Line 205: “i” should be in italics.

9. Page 8, Figure 6: Figure is not enough clear.

10. Page 8, Line 277: Why reference to Figure 6 is in parentheses?

11. Page 12, Figure 12: Please explain values X, Y in Figure 12b.

12. Page 12, Conclusion. The sentence “By fusing IMU and joint angle sensor, the slope orientation and inclination are estimated; according to the estimated slope information, the desired attitude and reference trajectory are modified so that the robot’s body is always parallel to the slope, leave sufficient motion space for front leg; the adjusted foot tochdown point keeps a large stability margin for CoM; the adaptive friction factor reduces the risk of foot slipping.“ is a bit complicated.

Reviewer 3 Report

First of all, congratulate the authors for the work carried out and the research paper sent to this journal for publication.

The authors in this paper describe an unknown slope-oriented research on model predictive control for a quadruped robot.

The abstract does not clearly indicate the contributions of the authors to the study problem addressed in the document. Some of these details appear throughout the document. In this way, the advances, results, conclusions, and importance of the research carried out are not well appreciated. It is necessary to indicate what is new in this paper. It is advisable to rewrite the summary indicating the main contributions, as well as the conclusions obtained.

The introduction incorporates a brief state-of-the-art. It is recommended that this state-of-the-art be much more detailed and include, for example, the main research studies and other problems associated with the analyzed problem. It may be interesting to make comparisons with other research papers already published, showing other techniques related to this research topic. In this case, the main contributions and novelties of the authors to the problem analyzed have not been very clear, nor have the advantages and disadvantages compared to other investigations carried out. As a suggestion, perhaps the elaboration of a comparative table that compiles the contributions to the study of the main bibliographical references, can give clarity to this state-of-the-art. The number of bibliographical references cited in the document is not adequate for the study carried out. It is necessary to increase the number of references to other similar works already published in the academic literature.

On the other hand, the document is technically solid, since it contains a brief analysis of the state-of-the-art (as has been commented, it is necessary to rewrite this section) related to the study developed. It also incorporates different concepts, details, and statistical information related to the development of the model. It also includes some estimated results that support the analysis shown.

The concepts have been presented exhaustively. The different figures, tables, diagrams, and attached diagrams facilitate the understanding of the contents presented by the authors in the document. Likewise, the simulated results obtained support the comments made by the authors. All this makes it easier for the reader to follow the different ideas presented in the document.

Most of the text incorporated in the conclusions section is more oriented to a discussion section than to the conclusions themselves. It would be advisable to rewrite this section and include, if necessary, a section for the discussion of the results. The conclusions section should include the main ideas, contributions, and results obtained by the authors in the study.

Finally, it is advisable to calmly read the English text or have a native English speaker proofread it, since words that are not commonly used in this language appear in the document.

Round 2

Reviewer 1 Report

Thank you for your edits and explanations in the article.

Reviewer 2 Report

Dear Authors,

The manuscript can be accepted for publication.

Reviewer